# Evaluation of the Cytotoxic, Anti-Inflammatory, and Immunomodulatory Effects of Withaferin A (WA) against Lipopolysaccharide (LPS)-Induced Inflammation in Immune Cells Derived from BALB/c Mice

**DOI:** 10.3390/pharmaceutics14061256

**Published:** 2022-06-13

**Authors:** Abdullah M. Alnuqaydan, Abdulmajeed Almutary, Gh Rasool Bhat, Tanveer Ahmad Mir, Shadil Ibrahim Wani, Mohd Younis Rather, Shabir Ahmad Mir, Bader Alshehri, Sulaiman Alnasser, Faten M. Ali Zainy, Bilal Rah

**Affiliations:** 1Department of Medical Biotechnology, College of Applied Medical Sciences, Qassim University, Buraidah 51452, Saudi Arabia; ami.alnuqaydan@qu.edu.sa (A.M.A.); abdulmajeed.almutary@qu.edu.sa (A.A.); 2Advanced Centre for Human Genetics, Sher-i-Kashmir Institute of Medical Sciences, Srinagar 190011, Jammu and Kashmir, India; seithbhat11@gmail.com (G.R.B.); shahdil.pu@gmail.com (S.I.W.); 3Laboratory of Tissue/Organ Bioengineering & BioMEMS, Organ Transplant Centre of Excellence, Transplantation Research & Innovation (Dpt)-R, King Faisal Specialist Hospital and Research Centre, MBC 03, Riyadh 11211, Saudi Arabia; mirtanveer0699@gmail.com; 4Multidisplinary Research Unit, Government Medical College, Srinagar 190010, Jammu and Kashmir, India; younisbiotech@gmail.com; 5Department of Medical Laboratory Sciences, College of Applied Medical Science, Majmaah University, Al Majmaah 11952, Saudi Arabia; mirshabir@gmail.com (S.A.M.); b.alshehri@mu.edu.sa (B.A.); 6Department of Pharmacology and Toxicology, Unaizah College of Pharmacy, Qassim University, Buraidah 51452, Saudi Arabia; sm.alnasser@qu.edu.sa; 7Chemistry Department, Faculty of Science, University of Jeddah, Jeddah 21589, Saudi Arabia; fmzainy@uj.edu.sa

**Keywords:** withaferin A, macrophages, immunosuppression, splenocytes, cytokines, inflammation

## Abstract

(1) Background: Inflammation is one of the primary responses of the immune system and plays a key role in the pathophysiology of various diseases. Recent reports suggest that various phytochemicals exhibit promising anti-inflammatory and immunomodulation activities with relatively few undesirable effects, thus offering a viable option to deal with inflammation and associated diseases. The current study evaluates the anti-inflammatory and immunomodulatory effects of withaferin A (WA) in immune cells extracted from BALB/c mice. (2) Methods: MTT assays were performed to assess the cell viability of splenocytes and anti-inflammatory doses of WA. Under aseptic conditions, the isolation of macrophages and splenocytes from BALB/c mice was performed to investigate the anti-inflammatory effects of WA. Analysis of the expression of proinflammatory cytokines and associated signaling mediators was performed using proinflammatory assay kits, real-time polymerase chain reaction (RT-PCR), and immunoblotting, while the quantification of B and T cells was performed by flow cytometry. (3) Results: Our results demonstrated that WA exhibits anti-inflammatory and immunomodulatory effects in LPS-stimulated macrophages and splenocytes derived from BALB/c mice, respectively. Mechanistically, we found that WA promotes an anti-inflammatory effect on LPS-stimulated macrophages by attenuating the secretion and expression of proinflammatory cytokines TNF-α, IL-1β, IL-6, and the inflammation modulator NO, both at the transcriptional and translational level, respectively. Further, WA inhibits LPS-stimulated inflammatory signaling by dephosphorylation of p-Akt-Ser473 and p-ERK1/2. This dephosphorylation does not allow IĸB-kinase activation to disrupt IĸB–NF-ĸB interaction. The consistent interaction of IĸB with NF-ĸB in WA-treated cells attenuates the activation of downstream inflammatory signaling mediators Cox-2 and iNOS expression, which play crucial roles in inflammatory signaling. Additionally, we observed significant immunomodulation of LPS-stimulated spleen-derived lymphocytes by suppression of B (CD19) and T (CD4^+^/CD8^+^) cell populations after treatment with WA. (4) Conclusion: WA exhibits anti-inflammatory and immunomodulatory activity by modulating Akt/ERK/NF-kB-mediated inflammatory signaling in macrophages and immunosuppression of B (CD19) and T cell (CD4^+^/CD8^+^) populations in splenocytes after LPS stimulation. These results suggest that WA could act as a potential anti-inflammatory/immunomodulatory molecule and support its use in the field of immunopharmacology to modulate immune system cells.

## 1. Introduction

Classically, inflammation is the interplay between host immune system components and infectious agents, autoimmune response, injury, or tissue ischemia [1]. The functional aspect of inflammation is either acute or chronic [2]. Acute inflammation is primarily associated with host defense mechanisms against pathogens or stimulation during the healing process. This type of inflammation is often beneficial and lasts for a few days to a few weeks, as in flu or bronchitis [3]. However, chronic inflammation is slow, without any stimulus, develops after acute inflammation is not resolved properly or with consistent exposure to chemicals and genetic susceptibility, and often remains for years [4]. With the advancement in scientific research, chronic inflammation has been documented to play a key role in the molecular mechanism of various chronic diseases, such as diabetes, cardiovascular diseases, atherosclerosis, cancer, inflammatory bowel diseases (IBD), neurodegenerative disorders, etc. [5]. Besides unresolved acute inflammation by infectious agents, the other key inducers of chronic inflammation are viruses and over-activation of the immune system [6]. The inflammatory response attracts innate as well as adaptive immune system cells towards the inflammation site, thereby initiating and maintaining the inflammation with the production of proinflammatory cytokines [7]. With the persistence of inflammation, both innate and adaptive immune cells attempt to engulf or destroy foreign bodies [8]. The unsuccessful attempts to clear the foreign agents lead to the formation of granulomas and often deregulate the proinflammatory cytokine production that orchestrates the underlying mechanism of chronic inflammation [9,10]. The major proinflammatory cytokines that are secreted and interact with cellular components of innate and adaptive immune cells, thereby promoting inflammation, are TNF-α, IL-6, IL-1β, IL-1, and NO, which is an inflammation modulator [11]. The excessive inflammatory response in the form of activation of immune cells and proinflammatory mediators often aggravates and causes tissue damage [12]. Therefore, modulation of the immune cells and proinflammatory cytokines in chronic diseases by anti-inflammatory drugs is the common approach to prevent further deterioration of tissues in vital organs such as the brain, kidneys, or liver [13]. Among the anti-inflammatory drugs, non-steroidal and corticosteroid drugs are often used to control pain and inflammation. However, owing to the various side effects, such as gastric ulcers and other deleterious effects caused by these drugs [14,15], the identification and evaluation of more specific and efficacious anti-inflammatory drugs is seemingly important to circumvent these issues.

Natural products are the prime source of inspiration for new small-molecule chemical entities in the field of drug discovery [16,17]. Natural products derived from plant sources are some of the major and important sources of existing as well as new drugs [17]. Recent reports suggest that the majority of pharmaceutical companies have decreased their focus on natural product drug discovery. Nonetheless, a small number of natural products and their leads have been tested in clinical trials as anticancer, antimicrobial, and antidiabetic drugs. A Food and Drug Administration report revealed that around 40% of approved medicines are either natural compounds or small molecules derived from natural compounds, including tubulin-interacted antitumor drugs, statins, and immunosuppressants [18]. Higher plant species comprise approximately 75,000 species on this planet, and not more than 10% have been used in the traditional medicinal system. Intriguingly, only up to 5% of these species have been explored scientifically with evaluations of their therapeutic potential owing to their photochemical constituents. This represents a vast reservoir of potential novel compounds [19]. Therefore, these plant species have been categorized as some of the prime and most extensive sources of natural compounds to be investigated for novel bioactivities.

One such natural compound derived from the winter cherry herb *Withania somenifera* (Ashwagandha), commonly found in the Himalayan regions of India, is withaferin A (WA) [20]. Chemically, a steroidal lactone, WA and its derivatives exhibit numerous health benefits and pharmacological activities, including anticancer [20,21,22,23], immunomodulatory, and anti-inflammatory potential [24]. Mechanistically, WA modifies the cellular morphology by modulating the cytoskeletal architecture by inhibiting proteasomal chymotrypsin-like activity and protein kinase C [25,26]. Besides the pharmacological activities mentioned above, WA also exhibits anti-inflammatory properties against various in vitro and in vivo models. WA restores IκB interaction with its binding partner NF-κB and prevents inflammation-associated signaling pathways by attenuating IκB kinase activation. Additionally, WA was reported to inhibit inducible nitric oxide synthase (iNOS) and NO production in Raw 264.7 cells after LPS sensitization [24]. In the current study, we performed a comprehensive anti-inflammation and immunomodulation evaluation of WA using immune cells (macrophages and splenocytes) isolated from BALB/C mice as cellular models. We demonstrated that WA not only reduces the secretion of proinflammatory mediators (TNF-α, IL-6, and IL-1β) and NO production, which acts as an inflammation modulator, but also attenuates the expression of proinflammatory cytokines at the transcriptional as well as the translational level. Further, to elucidate whether WA affects the NF-kB-mediated inflammatory signaling pathway, our results demonstrated that WA promotes dose-dependent dephosphorylation of LPS-stimulated NF-kB at p-p65 (Ser276, Ser536) with concomitant dephosphorylation of upstream p-ERK1/2, p-Akt (Ser473) and decreased expression of downstream signaling proteins Cox-2 and iNOS, respectively. Additionally, our study revealed that the dose-dependent treatment of WA promoted significant inhibition of CD4^+^, CD8^+^, and CD19^+^ cell populations in mouse-derived splenocytes after LPS + ConA stimulation. Together, the data suggest that WA exhibits promising anti-inflammation and immunomodulatory effects against well-established immune cells derived from the BALB/c model by inhibiting pro-inflammatory mediators and T (CD4^+^/CD8^+^) and B cell (CD19) proliferation. These results suggest that WA is a promising anti-inflammatory and immunomodulatory natural compound, though further preclinical studies are needed to authenticate its anti-inflammatory and immunomodulatory potential in the field of immunopharmacological translational medicine.

## 2. Materials and Methods

### 2.1. Chemicals, Reagents, Antibodies, and Assay Kits

E. coli O55 B5-derived LPS (#L2630), withaferin A (WA) (#W4394), concanavalin A (ConA) (#C5275), dimethyl sulphoxide (DMSO) (#C6164), streptomycin (#S9137), penicillin (#P4333), Hank’s balanced salt solution (HBSS) (#H1641), 3-4,5-diemthylthiazol-2yl-2,5-biphenyl 2,5-dimethyltetrazolium bromide (MTT) (#M5655), trypan blue (#T6146), fetal bovine serum (FBS) (#F2442), and RPMI-1640 (#R8758) were purchased from Sigma-Aldrich(St. Louis, MO, USA). All the primary antibodies, TNF-α (# 8184S), IL-6 (#12912S), IL-1β (#12703S), iNOS (#13120), p-p65 (Ser-276) (# 8242S), p-p65 (Ser-536) (#3033S), p-IkB-α (#2859S), IkB-α (# 9242S), ERK1/2 (#9194S), p-ERK1/2 (#9101S), Akt (#4885S), p-Akt (Ser473) (#4060S), Cox-2 (#4842S), and β-actin (#8457S) were procured from Cell Signaling Technology (Danvers, MA, USA). The secondary anti-mouse (#SC-2005) and anti-rabbit coupled with horseradish peroxidase antibodies (#SC-2357) were purchased from Santa Cruz Biotechnology (Dallas, TX, USA). The dilution factors for the primary and secondary antibodies were 1:1000 and 1:10,000, respectively. The cytokine assay kits were procured from R&D Systems (Minneapolis, MN, USA).

The current study was conducted on immune cells, such as macrophages and splenocytes, isolated from BALB/c (male) mice to evaluate the anti-inflammatory and immunomodulatory effects of WA. The immunomodulatory effect of WA was assessed as per the approved guidelines of the institutional animal ethics committee. Animals were bred and maintained as per the standard conditions: 12 h day–night photoperiod, 25 °C ± 2 °C temperatures. Animals were fed with a standard pellet diet and water ad libitum.

### 2.2. Isolation of Splenocytes (Lymphocytes) and Determination of Cell Viability

After collecting the spleens from the animals (BALB/c mice) under sterile conditions inside the laminar flow hood in HBSS, using scissors, the spleens were minced and allowed to pass through a steel mesh to achieve a homogenous suspension of cells, as described previously [27]. The cell suspension obtained containing red blood cells (RBCs) was lysed in a (0.8% *w*/*v*) ammonium chloride solution. The suspension of cells in ammonium chloride solution was centrifuged (380× *g*, 10 min at 4 °C), pelleted, washed with PBS (three times), and finally resuspended in RPMI-1640 supplemented with 10% FBS and 1% penicillin–streptomycin. Using trypan blue dye, cell counting was performed with a haemocytometer for the determination of cell viability, which should be more than 95%.

To determine the effect of withaferin A on the cell viability of lymphocytes derived from the spleens, briefly, 10^4^ cells were obtained from minced spleens and plated in each well of a 96-well plate. After incubation, plated cells were exposed to varying doses of WA along with LPS (1 μg/mL) and ConA (5 μg/mL) for 72 h. Later, each well was filled with 20 µL of MTT (2.5 mg/mL) and the plate was incubated in the incubator for 4 h. The plate was centrifuged (1400× *g*) for 5 min, and the supernatant was removed by pipetting carefully to avoid any loss of cells. Formazan crystals formed during the incubation of the cells with MTT were dissolved by adding 150 μL of DMSO and the plate was gently vortexed for 15 min to ensure complete mixing. Using an ELISA plate reader, absorbance was recorded at 570 nm and the data were processed to calculate percentage cell viability. For the validation of results, the experiment was repeated more than three times.

### 2.3. Collection of Peritoneal Macrophages and Estimation of Nitrite Content by Nitrite Assay

Under aseptic conditions, 10 mL of culture media (RPMI-1640) was loaded into a syringe and injected into the peritoneal cavity of BALB/c mice, as described previously [28]. After a period of 10 min, the medium mixed with peritoneal macrophages was taken out with the same syringe and centrifuged (1800× *g*) for 10 min at 4–8 °C. The cellular pellet thus obtained was resuspended in complete culture media (RPMI-1640).

Then, 3 × 10^6^ collected macrophages were plated in each well of a 24-well sterile culture plate and allowed to adhere to the bottom surface of the wells for 3 h in a 5% CO_2_ humidified incubator at 37 °C, as previously described [28]. After the incubation, the non-adherent cells were removed by aspirating the media from the wells, and fresh media was added with varying doses of WA along with LPS (1 μg/mL), and the cells were incubated for 48 h in a 5% CO_2_ humidified incubator at 37 °C. Afterward, the supernatant was collected from each well after centrifugation and stored at −80 °C for the estimation of cytokines and other inflammatory markers. For the determination of nitrite contents, 100 μL of culture media collected from each well was mixed with 100 μL of Griess reagent (2.5% phosphoric acid, 1% sulphanilamide, 0.1% naphthylethylenediamine) and incubated in a 96-well plate at 25 °C for 10 min. Using the ELISA plate reader, absorbance was recorded at 540 nm. The experiment was carried out in triplicate for the standard calibration curves and the validation of the results.

### 2.4. Evaluation of Pro-Inflammatory Cytokines (TNF-α, IL-6, and IL-1β) in Peritoneal Macrophages

As mentioned in Section 2.3, the collection of peritoneal macrophages was plated at a density of 3 × 10^6^ cells per well of a 6-well plate and treated with varying doses (0.25, 0.5, and 1.0 μM) of WA for 24 h, after stimulation with LPS (1 μg/mL) for 48 h, as described previously [29]. Afterward, the supernatant was collected and the estimation of cytokines TNF-α, IL-6, and IL-1β was performed as per the manufacturer’s instructions provided in the mouse TNF-α, IL-6, and IL-1β ELISA kits (R&D Systems, Minneapolis, MN, USA).

### 2.5. Immunophenotyping of Spleen-Derived Lymphocytes

As stated in Section 2.3, lymphocyte isolation was used to perform immunophenotyping according to the standard protocol [30]. Briefly, 3 × 10^6^ cells were exposed to varying doses of WA for 72 h in 5% humidified CO_2_ at 37 °C. Afterward, cells were processed, stained, and incubated with conjugated anti-CD19 PE, anti-CD8 FITC, and anti-CD4 PE antibodies for 30 min at 25 °C. After incubation, the stained cells were washed, resuspended, and processed in PBS for the flow cytometric analysis using the FACS Calibur flow cytometry system inbuilt in the Cell Quest software.

### 2.6. Immunoblotting Analysis

Immunoblotting was performed as per the standard protocol [31]. Following the aseptic collection of peritoneal macrophages from the BALB/c peritoneal cavity, the macrophages were resuspended in culture media and seeded at a density of 1 × 10^6^ cells in each well of a 6-well plate overnight in an incubator (5% CO_2_, 37 °C) to become properly attached to the bottom surface of the wells. The media was replaced with fresh media in the presence of LPS (1 μg/mL) alone or in combination with varying concentrations of WA for 48 h in a humidified incubator. The media was removed, and the cells were washed three times with ice-cold PBS before being harvested with a cell scraper. Hereafter, all the steps were performed on ice to avoid any protein degradation. This was followed by the addition of RIPA lysis buffer (500 μL) to each well, and cells were collected in 1.5 mL microcentrifuge tubes. The suspended cells in the lysis buffer were agitated for 3–5 s on the vortex every 5 min and put back on ice for 30 min. After centrifugation (10,956× *g*, 10 min at 4 °C), the supernatant obtained was collected in a separate microcentrifuge tube and protein estimation was performed by the Bradford method. After calculating the protein concentration in each sample, an equal amount of protein in the cell extract (30 μg) from each sample was loaded into each well of the SDS-PAGE gel. Proteins resolved by SDS-PAGE were transferred onto a PVDF membrane and non-specific antigen sites were blocked with blocking buffer (5% skimmed milk) for 1 h at room temperature. The PVDF membrane with primary antibodies was incubated for 3–4 h at room temperature or overnight at 4 °C, followed by gentle washing of the membrane with TBST buffer 3 times (5 min each time) and incubation with secondary antibodies coupled with horseradish peroxidase enzyme for 1–2 h at room temperature. The PVDF membrane was washed with TBST buffer three times (5 min each time). The membrane was allowed to semidry before the addition of chemiluminescent detection reagent for the detection of protein bands with a chemiluminescence gel doc detection system or with X-ray after development in the darkroom.

### 2.7. Real-Time (RT) Quantitative Reverse Transcription Polymerase Chain Reaction (PCR)

Under aseptic conditions, peritoneal macrophages collected from BALB/c mice were seeded at a density of 1 × 10^6^ cells in each well of a 6-well plate overnight in an incubator (5% CO_2_, 37 °C) to become attached properly to the bottom surface of the wells as per standard protocol [32]. Next, the media was replaced with fresh media in the presence of LPS (1 μg/mL) alone or in combination with varying concentrations of WA for 48 h in a humidified incubator. The total RNA extraction was performed using the TRIzol RNA extraction kit (Qiagen, Hilden, Germany). cDNA synthesis was performed with 2 μg total RNA, using the PrimeScript^TM^ RT Reagent Kit (Takara, Kusatsu, Japan). The cDNA obtained was used to conduct RT quantitative PCR for the genes with forward and reverse primer sequences as follows: iNOS forward 5′-TCCTACACCACACCAAAC-3′, reverse 5′-TCCTACACCACACCAAAC-3′; NF-kB (p65) forward 5′-GCGTACACATTCTGGGGAGT-3′, reverse 5′-CCGAAGCAGGAGCTATCAAC; TNF-α forward 5′-AGGTTCTGTCCCTTTCACTCACTGG-3′, reverse 5′-AGAGAACCTGGGAGTAGACAAGGTA-3′; IL-6 forward 5′-CCGGAGAGGAGACTTCACAG-3′, reverse 5′-TCCACGATTTCCCAGAGAAC-3′; IL-1β forward 5′- GAAGTCAAGAGAAAAGTGG-3′, reverse 5′-ACAGTCCAGCCCATACTTT-3′; and GAPGH forward 5′-TCAACGGCAAGTCAAGG-3′, reverse 5′-ACTCCACGACATACTCAG-3′, using SYBER^®^ Premix Ex Taq^TM^ and the Bio-Rad IQ5 Real-time PCR System (Bio-Rad, Hercules, CA, USA). Gene expression was analyzed after normalization with the loading control GAPDH expression.

### 2.8. Statistical Analysis

The experimental data obtained from the current study are presented as the mean ± SEM. The statistical analysis of the current study data was performed using one-way ANOVA (Bonferroni-corrected multiple comparison test). Dunnett’s *t*-test was used to examine the different variables in the same subject. A *p*-value equal to or less than 0.05 was considered statistically significant.

## 3. Results

### 3.1. The Effect of WA on Splenic Cell Viability and Splenocyte/Macrophage Proliferation Stimulated by ConA/LPS

The effect on cell viability of WA was determined to establish the appropriate dose for ongoing immunomodulatory and anti-inflammatory studies. A wide range of concentrations (0.125, 0.25, 0.5, 1.0, 2.0, 4.0, 8.0, 16.0 μM) of WA was selected to expose the splenocytes at different time points (48 and 72 h) to evaluate the cell viability of splenocytes. Our MTT assay results indicated a significant (approximately 60%, # *p* ≤ 0.05) viability of splenocytes when exposed to 2 μM or above after 48 h and 72 h compared to the untreated control. However, we observed a comparatively higher viability of splenocytes (≥79%, #*p* ≤ 0.05) at 1 μM or decreasing concentrations of WA (Figure 1) when exposed for 48 h and 72 h compared to the untreated control. These results indicate that low doses of WA (below 1 μM) are appropriate for ongoing anti-inflammation and immunomodulation studies.

To evaluate the effect of WA on cell-mediated immune response, we stimulated splenocytes and macrophages obtained from mouse spleens and peritoneal cavities with LPS (1 μg/mL) and ConA (5 μg/mL), respectively. After stimulation with mitogens (LPS/ConA), cultured splenocytes were exposed to varying doses of WA (0.25, 0.5, and 1.0 μM) for 72 h along with standard immunosuppressant betamethasone (BMS; 0.05 μM), which acts as a positive control, and LPS stimulation. The cell viability of spleen-derived lymphocytes and macrophages was determined by MTT assay. Our results indicated that WA significantly suppresses the cell viability of splenocytes and macrophages in a dose-dependent manner (# *p* ≤ 0.05), and the effect was most prominent with a higher dose (1.0 μM, # *p* ≤ 0.05) of WA. Additionally, we observed a significant inhibition of splenocyte proliferation and macrophages with BMS (0.05 μM, # *p* ≤ 0.05) as compared to LPS-stimulated splenocytes and macrophages, respectively (Figure 2A,B). These results indicate that WA exhibits promising immunomodulatory activity by inhibiting the proliferation of mitogen-stimulated splenocytes and macrophages derived from BALB/c mice.

### 3.2. WA Modulates Both the Expression and Secretion of Proinflammatory Cytokines in LPS-Stimulated Macrophages

The modulation and secretion of inflammatory cytokines by macrophages are the key hallmarks in the development of the pathophysiology of various diseases, including cancer and diabetes. To investigate whether WA could have any impact on the proinflammatory cytokine secretion of macrophages, we stimulated macrophages with LPS (1 μg/mL) and followed by the treatment with varying doses of WA (0.25, 0.5, and 1.0 μM), along with immunosuppressant BMS (0.05 μM) as positive control, LPS stimulation alone and untreated control group, for 48 h. After collecting conditioned media and using an ELISA assay kit (R&D systems) and a nitrite assay for the determination of NO, our results revealed that WA inhibits the secretion of TNF-α (Figure 3A), IL-6 (Figure 3B), IL-1β (Figure 3C), and nitrite concentration (Figure 3D) in a dose-dependent manner (# *p* ≤ 0.05, ## *p* ≤ 0.01). The inhibitory effect of cytokine secretion at 1.0 μM of WA was more pronounced and statistically more significant (## *p* ≤ 0.01) for all four cytokines. Additionally, the positive control BMS also showed a significant (# *p* ≤ 0.05) inhibitory effect on cytokine secretion from macrophages compared to the LPS-stimulated and untreated control groups.

To further investigate whether WA could attenuate the LPS-induced expression of pro-inflammatory cytokines at the translational and transcriptional levels, we exposed macrophages to different doses of WA (0.25, 0.5, and 1.0 μM), along with BMS (0.05 μM) and LPS stimulation for 48 h, in addition to an untreated control group. Our quantitative reverse transcription PCR results showed that WA downregulates the mRNA expression of all the four proinflammatory cytokines (TNF-α, IL-6, IL-1β, and iNOS) in a dose-dependent manner (# *p* ≤ 0.05, ## *p* ≤ 0.01). The immunosuppressant BMS was also found to downregulate the expression of TNF-α, IL-6, IL-1β, and iNOS significantly (# *p* ≤ 0.05) compared to LPS-stimulated and untreated control groups (Figure 4A). To evaluate the effect of WA on the expression of the proinflammatory cytokines at the translational level, our immunoblotting results revealed that upon dose-dependent treatment with WA after the LPS stimulation of macrophages, there was a sharp decrease in the protein expression of TNF-α, IL-6, IL-1β, and iNOS (Figure 4B). The effect of protein expression of TNF-α, IL-6, IL-1β, and iNOS was maximum at 1.0 μM of WA compared to the LPS-stimulated and untreated control groups (Figure 4C). Together, these results suggest that WA exhibits promising anti-inflammatory activity by downregulating proinflammatory cytokines TNF-α, IL-6, IL-1β, and iNOS both at transcriptional as well as translational levels after LPS stimulation. This downregulation led to a reduction in the secretion of proinflammatory cytokines from LPS-stimulated peritoneal macrophages isolated from BALB/c mice.

### 3.3. Effect of WA on the NF-ĸB-Mediated Inflammatory Signaling Pathway in LPS-Stimulated Macrophages

Among the inflammatory signaling pathways, NF-ĸB and associated signaling pathways play a major role in the activation of downstream mediators in inflammatory responses. To investigate whether WA modulates the expression of NF-ĸB in LPS-stimulated macrophages, we first sought to determine the expression of NF-ĸB in macrophages exposed to varying doses of WA, along with those in the BMS (0.05 μM), LPS-stimulated, and untreated control groups, after 48 h. Using RT-PCR and immunoblotting, we observed a significant dose-dependent downregulation of mRNA expression of p65 (Figure 5A; # *p* ≤ 0.05, ## *p* ≤ 0.01) with a concomitant reduction in the phosphorylation of p-p65 (Ser276, Ser536) in LPS-stimulated macrophages exposed to WA (Figure 5B,C; # *p* ≤ 0.05). To gain more insights into the underlying mechanism of WA-mediated regulation of NF-ĸB, we examined the impact of WA on the phosphorylation of IĸB, a binding partner of p65 that regulates the functionality of p65. As shown in Figure 5B,C, we observed a decrease in the IĸB phosphorylation of LPS-stimulated macrophages exposed to varying doses of WA (# *p* ≤ 0.05).

Accumulating evidence suggests that regulation of p65 phosphorylation at Ser 276 and Ser 536 is directly controlled by an extracellular signal-regulated kinase (ERK) and Akt, respectively. To investigate whether WA could also inhibit the expression of Akt, ERK, and associated signaling proteins involved in the activation of the inflammatory signaling pathway, we performed immunoblotting assays to analyze the expression of Akt/ERK and associated signaling proteins. Our immunoblotting results revealed that WA inhibits the expression of phosphorylation of Akt and ERK (Figure 6A,B). Intriguingly, we also observed that WA inhibits Cox-2 and iNOS expression in LPS-stimulated macrophages in a dose-dependent manner (Figure 6A,B; # *p* ≤ 0.05, ## *p* ≤ 0.01). The inhibitory effect on the phosphorylation status of p-Akt (Ser473), p-ERK1/2, and protein expression of COX-2 and iNOS was pronounced at 1.0 μM of WA compared to the untreated control (Figure 6A,B). Collectively, these data suggest that WA inhibits NF-ĸB-mediated inflammatory signaling by down-modulating the phosphorylation of upstream external signaling proteins p-Akt (Ser473) and p-ERK1/2, respectively. The dephosphorylation of upstream mediators (Akt, ERK1/2) by WA could not phosphorylate and release IĸB from its binding partner p65. The consistent interaction of p65 with its binding partner IĸB does not allow its nuclear translocation and the activation of downstream inflammatory mediators (Cox-2 and iNOS), promoting inflammatory signaling pathways in LPS-stimulated macrophages derived from BALB/c mice.

### 3.4. WA Modulates the Immunophenotyping of Spleen-Derived Lymphocytes from BALB/c Mice

Next, we intended to evaluate the immunomodulatory effect of WA on the splenocytes by quantification of the population of B and T cells with respective CD19, CD4, and CD8 cell surface markers using flow cytometry. After the LPS-stimulated splenocytes were exposed to different doses of WA (0.25, 0.5, and 1.0 μM), along with the immunosuppressant BMS (0.05 μM), LPS-stimulated, and untreated control groups, for 72 h, the flow cytometry analysis showed that significant (##*p* ≤ 0.01) (Table 1) inhibitory modulatory effects on both T cell subsets (CD4/CD8) (Figure 7A) and the B cell population (CD19) (Figure 7B) were observed when exposed to the dose-dependent treatment of WA. However, the immunomodulatory effect was more pronounced at a 1.0 μM dose of WA when compared to the positive control BMS (#*p* ≤ 0.05), LPS-stimulated, and untreated control groups. These results indicate that WA exhibits promising immunomodulatory activity against LPS-stimulated splenocytes derived from BALB/c mice.

## 4. Discussion

Inflammation and inflammatory responses are the synchronized stimulation of signaling pathways in immune cells that modulate the expression of inflammatory mediators [33]. The major players in the inflammatory responses associated with diabetes, obesity, atherosclerosis, and neurodegenerative diseases are immune components of both the adaptive and innate immune systems: macrophages, lymphocytes, and the associated proinflammatory cytokines [34]. Among the immune cells, activated macrophages are the major cellular components that secrete not only proinflammatory cytokines, such as NO, TNF-α, IL-1β, and IL-6, but also numerous toxic substances that cause shock, organ failure, fever, and help in the development of inflammation-associated pathophysiology [35]. The current regimen of drugs modulates these inflammatory responses and associated immune cells to avoid further tissue deterioration of some vital organs [36]. However, long-term use of these drugs has adverse side effects, such as immune system disturbances, and often culminates in severe conditions, such as immunosuppression and autoimmune disorders [37]. Therefore, a search for agents that regulate the fine balance of inflammation and associated immune cellular components in chronic diseases to avoid organ failure or further tissue damage with few or no side effects is urgently needed.

Accumulating evidence suggests that phytochemicals derived from plant sources exhibit anti-inflammatory potential with fewer deleterious side effects and offer a viable option for the therapeutics of inflammation-associated chronic diseases [38]. Due to their complex structural diversity, phytochemicals can modulate and target many inflammatory signaling pathways associated with proteins and immunomodulation associated with cellular components to prevent inflammation, with negligible side effects [39]. One such phytochemical derived from the medicinal plant *Withania somenifera* is WA. Chemically, a steroidal lactone that exhibits various pharmacological properties, WA is known to exhibit anti-inflammation-associated activities in different cellular models [40]. Here, we first demonstrated that WA exhibits dose-dependent (# *p* ≤ 0.05 at lower doses of WA; ## *p* ≤ 0.01 at higher doses of WA) LPS-stimulated anti-inflammatory and immunomodulatory potential in peritoneal macrophages and splenocytes derived from BALB/c mice. Mechanistically, we found that WA promotes an anti-inflammatory effect in LPS-stimulated macrophages by attenuating the secretion and expression of proinflammatory cytokines, such as NO, TNF-α, IL-1β, and IL-6, both at transcriptional and translational levels. Further, WA inhibits the phosphorylation of p-Akt (Ser473) and p-ERK1/2 and does not allow the phosphorylation and release of IĸB from its binding partner NF-ĸB (p65). The consistent interaction of NF-ĸB with IĸB in WA-treated cells attenuates the translocation and functionality of p65 to activate the downstream mediators of inflammatory signaling pathways, such as Cox-2 and iNOS. Additionally, we observed significant immunomodulation of LPS-stimulated spleen-derived B and T cell populations after dose-dependent treatment with WA. Together, these results suggest that WA exhibits promising anti-inflammatory and immunomodulatory activity in macrophages and splenocytes derived from BALB/c mice, respectively, and document its role in the field of immunopharmacology.

Previous studies have documented that WA has promising antiproliferative activity against a wide variety of cancer cell lines [26]. Therefore, in the present study, we first checked the cell viability of splenocytes derived from BALB/c mice to evaluate the safe dosage of WA for investigating its anti-inflammatory and immunomodulatory potential. Using a wide range of WA doses (0.125 to 16.0 μM) to expose cells for 72 h, our cell viability results by MTT assay suggest that 1.0 μM of WA has more than 70% cell viability and exhibits less than 30% toxicity to splenocytes. However, we observed less than 50% cell viability of splenocytes when exposed to 2.0 μM of WA for 72 h. Additionally, to decrease the cell-toxicity effect of WA, we shortened the exposure time of splenocytes to WA by 48 h to nullify any significant toxicity in upcoming experiments. Our results indicated that up to 1.0 μM of WA for 48 h has significant cell viability (# *p* ≤ 0.05) and is safe to investigate the anti-inflammatory and immunomodulatory effects in splenocytes isolated from BALB/c mice. Numerous reports suggest that natural compounds immunomodulate the cell-mediated immune response either by stimulating or suppressing the cell proliferation of lymphocytes to affect the humoral or cell-mediated immune response [41]. After selecting the safe dose of WA for anti-inflammatory and immunomodulatory activity (0.25, 0.50, 1.0 μM), we evaluated the cellular immune response of splenocytes after LPS and ConA stimulation for 72 h. Our results suggest that WA promotes the suppression of splenocyte proliferation in a dose-dependent manner.

Macrophages are the key players in inflammation-associated immune responses and play a critical role in the development of the pathophysiology of chronic diseases [11]. The activating macrophages not only secrete some major pro-inflammatory mediators, such as TNF-α, IL-6, NO, and IL-1β, but also release various biologically active toxic substances that participate in fever, anti-infections, organ failure, shock, and associated pathophysiological processes [42]. The classical pro-inflammatory cytokine, TNF-α, is mainly secreted by macrophages and has a major role in promoting acute lung injury [43]. Upregulation of TNF-α is a major hallmark in the pathophysiology of chronic inflammatory diseases and targeting TNF-α with anti-TNF-α agents is an ideal strategy to manage chronic inflammatory diseases [44]. IL-1β is another prominent proinflammatory cytokine and its upregulation is implicated in autoimmune disorders, neurologic diseases, vascular diseases, inflammatory bowel diseases, and multiple sclerosis [45]. Therefore, targeting IL-1β to attenuate its upregulation is another key therapeutic option to control chronic diseases. Moreover, another pro-inflammatory cytokine, IL-6, is secreted mainly by macrophages, and its deregulation is implicated in rheumatoid arthritis and inflammatory bowel disease [46]. Thus, regulating IL-6 expression is considered an important therapeutic option to manage these diseases. In the current study, we found that WA not only reduces the secretion of TNF-α, IL-6, and IL-1β in a dose-dependent manner but also downregulates their expression both at transcriptional and translational levels in LPS-stimulated peritoneal macrophages isolated from BALB/c mice. This indicates that the reduction in the secretion of these proinflammatory cytokines in LPS-stimulated macrophages is due to downregulation at the gene level by WA treatment.

NO is a key inflammatory modulator and oxidative mediator upregulated in various pathophysiological states of chronic inflammatory diseases [47]. Under normal circumstances, NO production is regulated by constitutive NOS, whereas in pathophysiological states of chronic inflammatory diseases, NO production is under the influence of inducible NOS (iNOS) [48]. NO production is also induced by immunological stimuli, such as IFN-γ or LPS, from macrophages and often culminates in myocardial ischemia, cerebral injury, septic shock, and diabetes [49]. Therefore, targeting NO production by natural compounds is potentially beneficial. In the present study, we found that WA promoted the attenuation of iNOS mRNA expression as well as protein expression of iNOS in a dose-dependent manner (# *p* ≤ 0.05 at lower doses of WA; ## *p* ≤ 0.01 at higher doses of WA) after the LPS stimulation of macrophages isolated from the peritoneal cavity of BALB/c mice. This attenuation of iNOS resulted in the downregulation of iNOS protein, which in turn reduced NO production in the LPS-stimulated macrophages exposed to WA treatments.

To further investigate the underlying inhibitory mechanism of inflammatory signaling pathways by WA in LPS-stimulated macrophages, we evaluated the effect of WA on upstream signaling mediators, such as extracellular signal-regulated kinase (ERK) and Akt, which directly regulate NF-ĸB and associated downstream mediators involved in inflammation. Previous reports suggest that extracellular signal-regulated kinase (ERK) and PI3K/Akt signaling have a major role in transducing NF-ĸB-mediated inflammatory signaling in macrophages [50]. Mechanistically, these upstream mediators phosphorylate IĸB through IĸB-kinase, thereby releasing, phosphorylating (Ser276 and Ser536), and translocating the activated p65 into the nucleus to act as a transcriptional factor for various genes, including Cox-2, iNOS, and associated proteins [51]. In the current study, we observed that WA inhibited the expression of phosphorylated Akt and ERK 1/2, with a concomitant decrease in the expression of Cox-2 and iNOS expression in LPS-stimulated macrophages in a dose-dependent manner. The transcriptional factor NF-ĸB plays a central role in the inflammatory signaling pathways by interacting with the promoter regions of numerous genes that are involved in inflammation-associated diseases [52,53]. Under normal physiological conditions in unstimulated cells, NF-ĸB is localized in the cytosol as homodimers or heterodimers with its inhibitory binding partner, IĸB, whereas in stimulated cells, the phosphorylation of the inhibitory binding partner IĸB breaks the heterodimer interaction between IĸB and NF-ĸB. This dissociation allows NF-ĸB to translocate into the nucleus to act as a transcriptional factor by interacting with consensus sequences in the promoter regions of numerous genes involved in inflammatory responses [50]. Previous reports suggest that various natural compounds exhibit an anti-inflammatory effect by inhibiting IĸB kinase activity. This attenuation of IĸB kinase activity sustains the interaction of IĸB with NF-ĸB and does not allow NF-ĸB to translocate into the nucleus to perform its function as a transcriptional factor for various inflammatory genes [53,54,55]. Consistent with previous reports, we found a dose-dependent downregulation of p65 mRNA (# *p* ≤ 0.05 at lower doses of WA; ## *p* ≤ 0.01 at higher doses of WA) as well as phosphorylation of p-p65 (Ser276, Ser536) protein expression with WA treatment in LPS-stimulated macrophages (# *p* ≤ 0.05). Further, we observed a decrease in IĸB phosphorylation in LPS-stimulated macrophages exposed to varying doses of WA, which indicates that WA might inhibit IĸB kinase activity, which in turn means that IĸB is not able to phosphorylate and release IĸB from its binding partner NF-ĸB to inhibit the downstream signaling of inflammatory proteins. Collectively, these data suggest that WA inhibits the dephosphorylation of upstream mediators (Akt, ERK1/2). This dephosphorylation could not activate IĸB kinase, thereby maintaining the interaction of IĸB with its binding partner p65. The consistent interaction of p65 with its binding partner IĸB does not allow the activation and translocation of p65 into the nucleus to promote the expression of downstream inflammatory mediators, such as Cox-2 and iNOS, and thus attenuated the pro-inflammatory signaling pathway in LPS-stimulated macrophages derived from BALB/c mice.

The modulation of immune response in terms of cellular and humoral components of the immune system after the administration of a drug or natural compounds into the body is called immunomodulation [56]. This modulation includes inhibition (immunosuppression), induction (via immunostimulators), or amplification of various phases of the immune response [57]. For various immune-associated disorders, such as autoimmune disorders, cancer, or diseases such as tuberculosis or AIDS, the host defense mechanism is impaired and modulation is needed to boost the immune system to combat such diseases [58]. Natural compounds obtained from plant sources have been documented to have immunomodulatory effects on T lymphocytes, which are the major components of cellular immunity [59]. The T cells (T helper cells and cytotoxic T lymphocytes) not only immunopotentiate CD4^+^ and CD8^+^ Tlymphocytes but also enhance the proliferation, maturation, and antigenic sensitization of other cell types. Further, T helper cells specifically secrete some chemokines which activate macrophages, cytotoxic T lymphocytes, and B cells [60]. The cytotoxic T lymphocytes provide immunity against viral infections; however, B cells are the major contributors to humoral immunity and express 19 clusters of differentiation (CD) antigens on the cell surface that play a critical role in B cell proliferation [61]. In the current study, we evaluated the immunomodulatory effect of WA in LPS-stimulated splenocytes. Our flow cytometry analysis revealed that WA inhibits the cell population of CD4^+^ and CD8^+^ T lymphocytes in a dose-dependent manner (## *p* ≤ 0.01 at all doses of WA) when LPS-stimulated splenocytes are exposed to varying doses of WA. Additionally, we also observed a significant decrease in the CD19 (B cell) population when exposed to varying doses of WA. These results suggest that WA immunosuppresses the spleen-derived LPS-stimulated T and B cell population in a dose-dependent manner (## *p* ≤ 0.01) and therefore enables an immunomodulatory effect. 

## 5. Conclusions

The current study revealed that WA induces an anti-inflammatory effect on LPS-stimulated macrophages by attenuating proinflammatory cytokines both at the transcriptional and translational levels. Further, we observed that WA inhibits NF-ĸB-mediated attenuation of the inflammatory signaling pathway by decreasing the phosphorylated form of upstream signaling proteins p-ERK1/2 and p-Akt, which in turn leads to the dephosphorylation of p-p65 (Ser276, Ser536) and p-IĸB-α, with a concomitant reduction in the expression of Cox-2 and iNOS proteins. Additionally, we observed dose-dependent immunomodulation (a decrease the cell populations of CD4^+^, CD8^+^, and CD19) in LPS-stimulated splenocytes (Figure 8). These data suggest that WA exhibits promising anti-inflammatory/immunomodulatory effects, though further evaluation is needed at the preclinical stage to validate immunopharmacological applications before launching clinical trials for various immunological disorders.

## Figures and Tables

**Figure 1 pharmaceutics-14-01256-f001:**
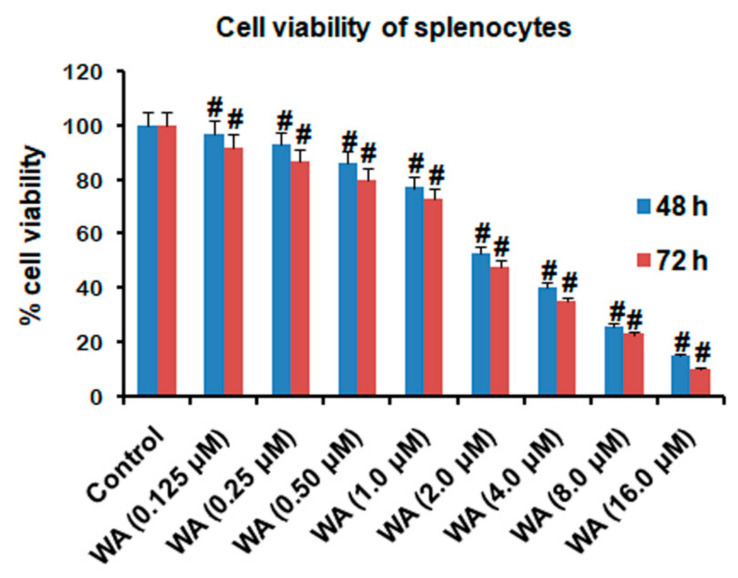
Cell viability of splenocytes isolated from BALB/c mice. After being seeded in a 96-well plate at a density of 10^4^ cells per well, splenocytes were exposed to a range of doses of WA (0.125, 0.25, 0.5, 1.0, 2.0, 4.0, 8.0, 16.0 μM) along with untreated splenocytes for 48 and 72 h to evaluate the cell viability of splenocytes after treatment with WA. Experimental data are presented as the mean ± SEM from three independent experiments. # *p* ≤ 0.05 is statistically significant.

**Figure 2 pharmaceutics-14-01256-f002:**
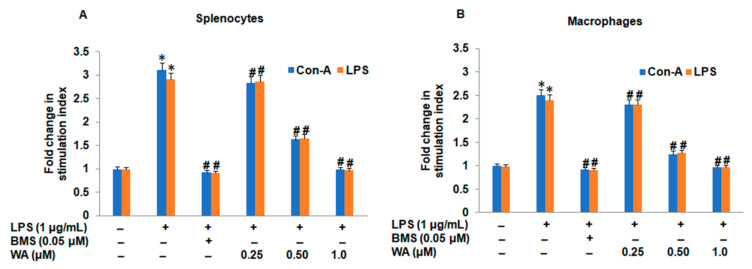
WA inhibits ConA/LPS-stimulated splenocytes and macrophage cell viability. (**A**) Isolated splenocytes were initially stimulated with mitogens LPS (1 μg/mL) and Con5 (5 μg/mL) to enhance the proliferation of spleen-derived lymphocytes (B and T cells) followed by treatment with varying doses of WA (0.25, 0.5, and 1.0 μM) for 72 h along with positive control BMS (0.05 μM) and LPS-stimulating splenocytes to evaluate cell viability. (**B**) Macrophages collected from the peritoneal cavity of mice were initially stimulated with LPS (1μg/mL) and Con5 (5 μg/mL) mitogens to enhance their proliferation, followed by treatment with varying doses of WA (0.25, 0.5, and 1.0 μM) for 72 h along with positive control BMS (0.05 μM) and LPS-stimulating macrophages to evaluate cell viability. Experimental data are presented as the mean ± SEM of three independent experiments. * *p* ≤ 0.05 (comparison vs. LPS only) and # *p* ≤ 0.05 (comparison vs. +LPS) are statistically significant.

**Figure 3 pharmaceutics-14-01256-f003:**
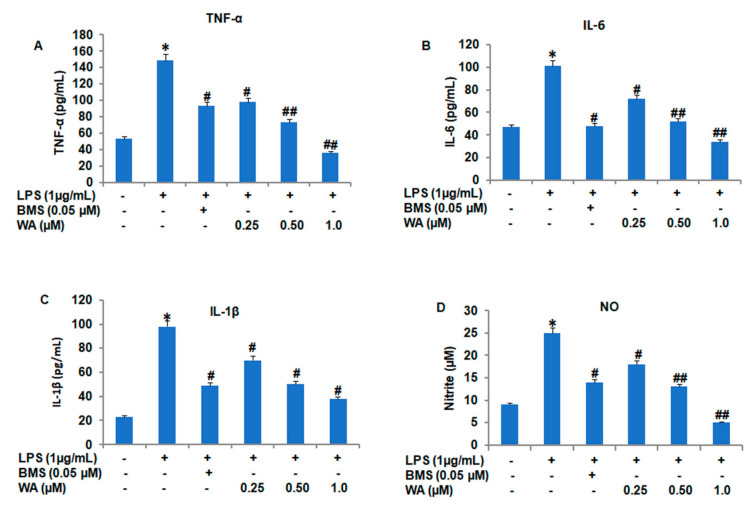
WA inhibits pro-inflammatory cytokines secreted by macrophages isolated from the peritoneal cavity of BALB/c mice. (**A**–**D**) The histogram represents the level of TNF-α (pg/mL), IL-6 (pg/mL), IL-1β (pg/mL), and NO (µg/mL) in conditioned media obtained from cultured macrophages after LPS stimulation, followed by exposure to different doses (0.25, 0.5, and 1.0 μM) of WA, along with BMS (0.05 μM) and LPS stimulation alone and untreated control group, after 48 h. Experimental data are presented as the mean ± SEM from three or more independent experiments. * *p* ≤ 0.05 (comparison vs. LPS only) and # *p* ≤ 0.05, and ## *p* ≤ 0.01 (comparison vs. +LPS) are statistically significant.

**Figure 4 pharmaceutics-14-01256-f004:**
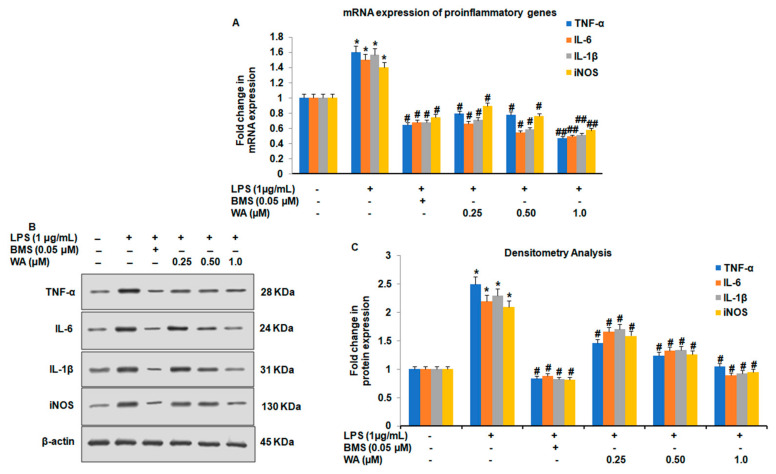
Effect of WA on the expression of pro-inflammatory cytokines after LPS stimulation in peritoneal macrophages. (**A**) The histogram represents the mRNA expression by RT-quantitative PCR of proinflammatory cytokines TNF-α, IL-6, IL-1β, and iNOS of LPS (1 μg/mL)-stimulated peritoneal macrophages exposed to varying doses of WA, along with the BMS (0.05 μM), LPS-stimulated, and untreated control groups, after 48 h. (**B**) Protein expression as determined by immunoblotting of pro-inflammatory cytokines TNF-α, IL-6, IL-1β, and iNOS of LPS (1 μg/mL)-stimulated peritoneal macrophages exposed to varying doses of WA, along with the BMS (0.05 μM), LPS-stimulated, and untreated control groups, after 48 h. (**C**) The histogram represents the fold changes in protein expression as determined by densitometry of pro-inflammatory cytokines TNF-α, IL-6, IL-1β, and iNOS of LPS (1 μg/mL)-stimulated peritoneal macrophages exposed to varying doses of WA, along with BMS (0.05 μM), LPS-stimulated, and untreated control groups, after 48 h. Experimental data are presented as the mean ± SEM of three independent experiments. * *p* ≤ 0.05 (comparison vs. LPS only), # *p* ≤ 0.05, and ## *p* ≤ 0.01 (comparison vs. +LPS) are statistically significant.

**Figure 5 pharmaceutics-14-01256-f005:**
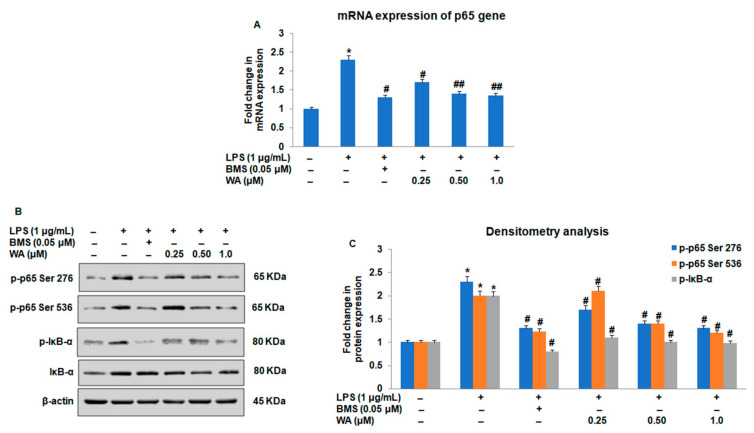
Effect of WA on the LPS-stimulated inflammatory signaling pathway in peritoneal macrophages. (**A**) Histograms represent the mRNA expression as determined by RT-quantitative PCR of NF-ĸB (p65) in LPS (1 μg/mL)-stimulated peritoneal macrophages exposed to varying doses of WA, along with the BMS (0.05 μM), LPS-stimulated, and untreated control groups, after 48 h. (**B**) Protein expression and phosphorylation status by immunoblotting of p-65 and IĸB of LPS (1 μg/mL)-stimulated peritoneal macrophages exposed to varying doses of WA, along with the BMS (0.05 μM), LPS-stimulated, and untreated control groups, after 48 h. (**C**) Histogram represents fold changes in protein expression as determined by densitometry of p-65 and IĸB of LPS (1 μg/mL)-stimulated peritoneal macrophages exposed to varying doses of WA, along with the BMS (0.05 μM), LPS-stimulated, and untreated control groups, after 48 h. Experimental data are presented as the mean ± SEM of three or more independent experiments. * *p* ≤ 0.05 (comparison vs. LPS only), # *p* ≤ 0.05, and ## *p* ≤ 0.01 (comparison vs. +LPS) are statistically significant.

**Figure 6 pharmaceutics-14-01256-f006:**
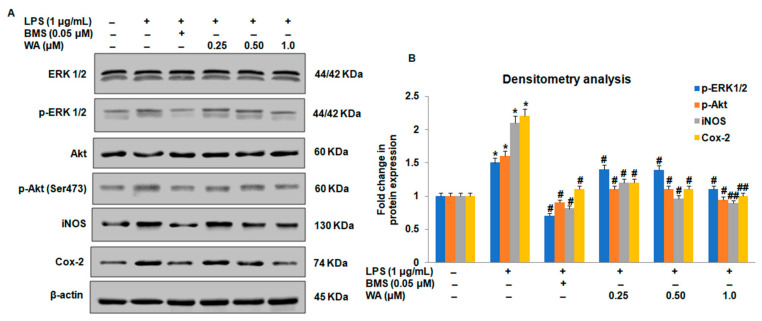
Effect of WA on upstream Akt-ERK1/2 and downstream mediators of the NF-kB-mediated signaling pathway in LPS-stimulated macrophages. (**A**) Protein expression and phosphorylation status by immunoblotting of Akt, ERK1/2, Cox-2, and iNOS of LPS (1 μg/mL)-stimulated peritoneal macrophages exposed to varying doses of WA, along with the BMS (0.05 μM), LPS-stimulated, and untreated control groups, after 48 h. (**B**) The histogram represents the fold changes in protein expression as determined by densitometry of Akt, ERK1/2, Cox-2, and iNOS of LPS (1 μg/mL)-stimulated peritoneal macrophages exposed to varying doses of WA, along with the BMS (0.05 μM), LPS-stimulated, and untreated control groups, after 48 h. Experimental data are presented as the mean ± SEM of three or more independent experiments. * *p* ≤ 0.05 (comparison vs. LPS only), # *p* ≤ 0.05, and ## *p* ≤ 0.01 (comparison vs. +LPS) are statistically significant.

**Figure 7 pharmaceutics-14-01256-f007:**
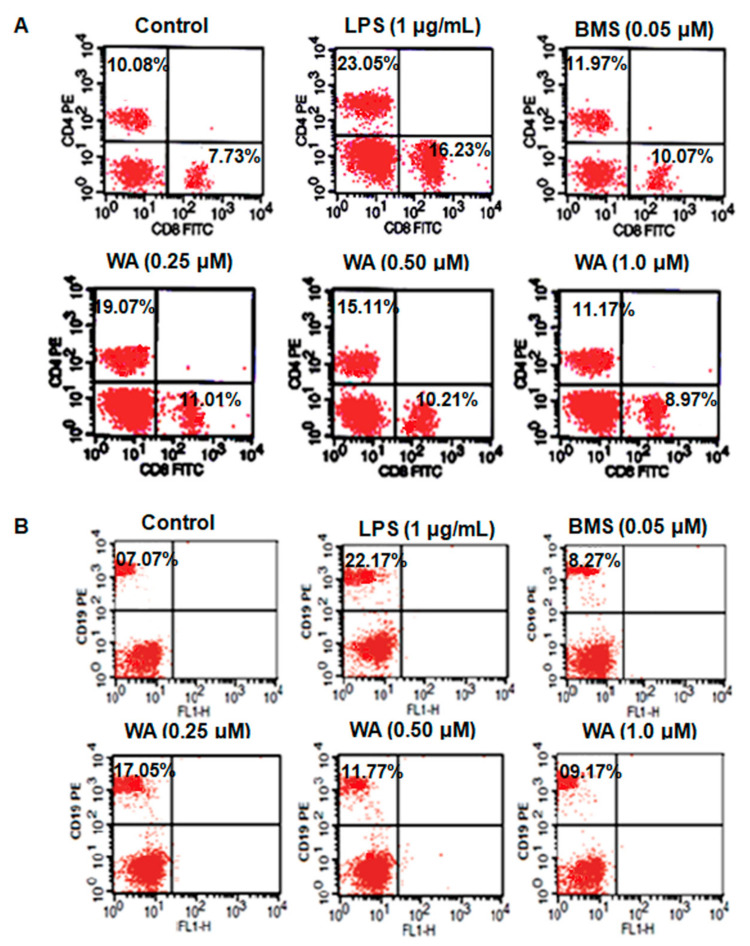
WA immunomodulates the LPS-stimulated splenocytes derived from BALB/c mice. (**A**) Quantification of T cell subsets (CD4/CD8) by flow cytometry of LPS/ConA-stimulated splenocytes exposed to indicated doses of WA and labeled with surface antibody markers CD4 (PE-coupled monoclonal antibody) and CD8 (FITC-coupled monoclonal antibody). (**B**) Quantification of B cells (CD19) by flow cytometry of LPS-stimulated splenocytes exposed to indicated doses of WA and labeled with surface antibody markers CD19 (PE-coupled monoclonal antibodies).

**Figure 8 pharmaceutics-14-01256-f008:**
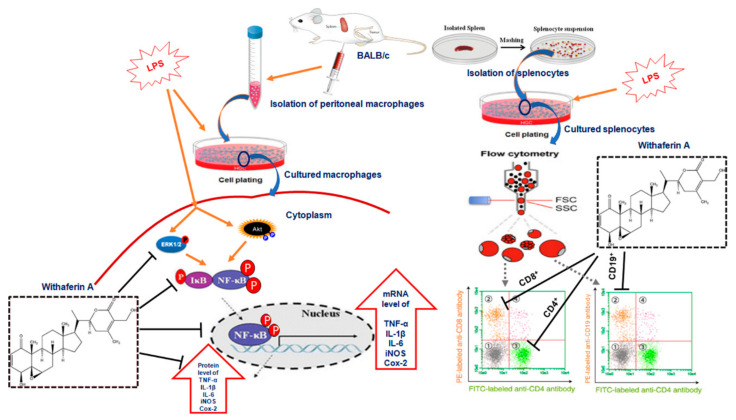
The schematic diagram represents how WA induces an anti-inflammatory effect and immunomodulates B and T cell populations in macrophages and splenocytes derived from BALB/c mice.

**Table 1 pharmaceutics-14-01256-t001:** Immunomodulatory effect of WA on LPS-stimulated splenocytes derived from BALB/c mice. Experimental data presented as the mean ± SEM of three or more independent experiments. * *p* ≤ 0.05 (comparison vs. LPS only), # *p* ≤ 0.05, and ## *p* ≤ 0.01 (comparison vs. + LPS) are statistically significant.

Treatment	Dose (µM)	% CD4^+^	% CD8^+^	% CD19^+^
Control	-	10.08 ± 1.04	7.73 ± 0.98	7.07± 0.88
LPS	(1 μg/mL)	23.05 ± 1.24 *	16.23 ± 1.4 *	22.17 ± 1.23 *
BMS	0.05	11.97 ± 0.13 ^#^	10.07 ± 0.17 ^#^	8.27 ± 0.4 ^#^
WA	0.25	19.07 ± 0.18 ^##^	11.01 ± 0.11 ^##^	17.05 ± 0.14 ^##^
WA	0.50	15.11 ± 0.12 ^##^	10.21 ± 0.16 ^##^	11.77 ± 0.09 ^##^
WA	1.0	11.97 ± 0.11 ^##^	8.97 ± 0.22 ^##^	9.17 ± 0.13 ^##^

## Data Availability

The data used to support the findings of this study are available from the corresponding author upon request.

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
