# Peer review of "Evaluation of the Cytotoxic, Anti-Inflammatory, and Immunomodulatory Effects of Withaferin A (WA) against Lipopolysaccharide (LPS)-Induced Inflammation in Immune Cells Derived from BALB/c Mice"

_pharmaceutics, 2022, doi:10.3390/pharmaceutics14061256_

Round 1
Reviewer 1 Report
The manuscript 1748316 by Alnuqaydan et al. aims to evaluate the effects of withaferin A on LPS-induced inflammation in murine macrophages and splenocytes. They found that withaferin A is able to attenuate secretion and expression of proinflammatory cytokines and generation of NO, and they provide information on the mechanism by analyzing the expression of Cox-2 and iNOS and of phosphorylated AKT, ERK, and NFkB . The results are interesting, and the manuscript is fairly written, but unfortunately, the description of the statistical analysis of the experiments is unclear, or it is missing, and therefore, it is not possible to conclude as stated in the abstract, in the results and in the conclusions that there is a dose-dependent effect of withaferin A. Other points that should be corrected are the interpretation of the MTT assay, which does not measure cell death; NO is a modulator of inflammation, but it is not a cytokine.
291: Our MTT assay results indicated that a significant (> 50 %) population of splenocytes 291 were killed when exposed to 2 μM or above after 48 h compared with 72 h. MTT assay is a viability test; it is not possible to distinguish between arrest of cell growth and cell death. Therefore, it is not correct to write that more than 50% of cells were killed, only that viability was 50% compared to control. Similarly, it is not correct “comparatively less cell killing of splenocytes at 1 μM or low doses of WA”, the results indicate that viability of splenocytes was higherat decreasing WA concentrations. However, the statistical analysis that allows to state this is not included in the figure or in the text. Apparently, # indicates the significance versus control cells. Other symbols should be used to indicate the statistical differences between different concentrations of WA .
FIGURE 2:
The statistical analyses of Figures 2 A and B are not clear. Please specify whether # refers to the different combinations of WA + LPS, BMS - LPS versus LPS alone
There is a discrepancy between text and legend. Maybe Con-5 (1 μg/mL) should be substituted with Con-A (5 μg/mL). Please check.
Please correct spenocytes in the title of Figure 2A
337-338 our results revealed that WA inhibits the secretion of TNF-α (Figure 3A), IL-6 (Figure 3B), IL-1β (Figure 3C), and nitrite concentration (Figure 3D) in a dose-dependent manner.
In order to demonstrate that the inhibition is dose-dependent, the p values of the comparisons between the different concentrations of WA should be included, and different symbols or bars linking the histograms should be shown. Again, it is not clear the meaning of # (comparison versus presence of only LPS?)
341 Additionally, the positive control BMS also showed a significant (# ≤ 0.05) inhibitory effect on cytokine secretion from macrophages compared to the LPS-stimulated and untreated control groups .
I suggest to use two different symbols, for example # to indicate the comparison vs + LPS and * to indicate the comparison versus control (- LPS)
Figure 4A,B:
Again, I think that the results should be clearer using different symbols to indicate the statistical significance versus control (* for example) and # versus LPS. Again, if the Authors want to demonstrate that there is a dose-dependent effect, another symbol (for example,§) should be used in the figures to indicate the difference between the higher and the lower WA concentrations, or the information regarding the p values should be added in the text.
Figure 5A,B; Figure 6 B
The Authors should make clear which are the comparisons made, and indicate whether # means versus +LPS or versus –LPS, and use two different symbols to indicate the statistical significance versus +LPS and versus –LPS.
430-431 Intriguingly, we also observe that WA inhibits Cox-2 and iNOS expression in LPS-stimulated macrophages in a dose-dependent manner (Figure 6 A, B).
It is not clear whether there is a significant dose-dependence in the inhibitory effect of WA on Cox-2 and iNOS expression. To demonstrate that, p values of the comparison between the different concentrations of WA should be included, at least in the text.
505-506 Here, we first demonstrated that WA exhibits dose-dependent LPS-stimulated anti-inflammatory and immunomodulation potential on the peritoneal macrophages and splenocytes derived from BALB/c mice.
There is not statistical significance analyses demonstrating that the effect is dose-dependent, that is, that the effect of the lower dose is different from the effect of the higher dose. The trend is clear from the histograms, but the information on the statistics is missing.
509 proinflammatory cytokines like NO, TNF-α, IL-1β, and IL-6 both at transcriptional and translational levels.
NO is not a cytokine, although it is a mediator of inflammation.
526: However, we observed more than 50 % cell killing when splenocytes were exposed to 2.0 μM of WA for 72 h. See comment above. Killing should be avoided.
546 Upregulation of TNF-α is a major hallmark in the pathophysiology of chronic inflammatory diseases is anti-TNF-α therapy [44]. This sentence is unclear
567-569: we found WA promotes attenuation of iNOS mRNA expression as well as protein expression of iNOS in a dose-dependent manner after LPS-stimulation of macrophages isolated from the peritoneal cavity of BALB/c mice.
See above: to state this it is necessary to add the statistical analyses of the different effect between at least two doses of WA
598-599 Consistent with previous reports, we found a dose-dependent downregulation of p65 mRNA as well as phosphorylation of p-p65 (Ser276, Ser536) protein expression with WA treatment in LPS-stimulated macrophages.
633-634 These results suggest that WA immunosuppresses the spleen-derived LPS-stimulated T and B-cell population in a dose-dependent manner and therefore enables its immunomodulatory effect.
Although I agree that different doses of WA downregulate mRNA, the Authors have not demonstrate a dose-dependent downregulation (see above). It would be necessary to add the statistical analysis showing that there is difference when comparing at least two doses of WA.
Minot points:
Some sentences are not clear and need to be rewritten:
23 Inflammation is one of the primary responses and plays a key role in the pathophysiology of various diseases
The sentence seems incomplete. Primary response to what?
33 However, the quantification of B and T cells was performed and flow cytometry.
111 By specifically inhibiting proteasomal chymotrypsin-like activity and protein kinase C [26].
38 NO is not a cytokine
Some typos should be corrected:
44 by suppresses
98 immunosuppressant’s [18].
187 with 20 L of MTT The volume unit is wrong. I guess that 20 microL have been added.
188 Centrifuge the plate (1400 x g) for 5min Please rewrite. The plate was centrifuged or after centrifugation…
385 histograms represents
Author Response
Reviewer 1:
Comment 1: The manuscript 1748316 by Alnuqaydan et al. aims to evaluate the effects of withaferin A on LPS-induced inflammation in murine macrophages and splenocytes. They found that withaferin A is able to attenuate secretion and expression of proinflammatory cytokines and generation of NO, and they provide information on the mechanism by analyzing the expression of Cox-2 and iNOS and of phosphorylated AKT, ERK, and NFkB . The results are interesting, and the manuscript is fairly written, but unfortunately, the description of the statistical analysis of the experiments is unclear, or it is missing, and therefore, it is not possible to conclude as stated in the abstract, in the results and in the conclusions that there is a dose-dependent effect of withaferin A. Other points that should be corrected are the interpretation of the MTT assay, which does not measure cell death; NO is a modulator of inflammation, but it is not a cytokine.
Reply to comment 1: Thank you so much for critically reviewing our manuscript and appreciated writing skills. We have performed a statistical analysis of all the results discussed in the manuscript. As suggested by the reviewer, we have incorporated the values as well as statistical analysis in the manuscript. We have edited the few sentences in the abstract section to make them more meaningful and corrected other typo errors.
Comment 2: Our MTT assay results indicated that a significant (> 50 %) population of splenocytes 291 were killed when exposed to 2 μM or above after 48 h compared with 72 h. MTT assay is a viability test; it is not possible to distinguish between arrest of cell growth and cell death. Therefore, it is not correct to write that more than 50% of cells were killed, only that viability was 50% compared to control. Similarly, it is not correct “comparatively less cell killing of splenocytes at 1 μM or low doses of WA”, the results indicate that viability of splenocytes was higherat decreasing WA concentrations. However, the statistical analysis that allows to state this is not included in the figure or in the text. Apparently, # indicates the significance versus control cells. Other symbols should be used to indicate the statistical differences between different concentrations of WA .
Reply to comment 2: As suggested by the reviewer, we have edited the above-mentioned sentences in the result section under the subheadings 3.1 of the manuscript. The incorporated text is highlighted in red color text. The statistical analysis performed is incorporated in Figure 1.
Comment 3: FIGURE 2:
The statistical analyses of Figures 2 A and B are not clear. Please specify whether # refers to the different combinations of WA + LPS, BMS - LPS versus LPS alone
There is a discrepancy between text and legend. Maybe Con-5 (1 μg/mL) should be substituted with Con-A (5 μg/mL). Please check.
Please correct spenocytes in the title of Figure 2A
Reply to comment 3: Thanks for the critical revision which helps us to improve our manuscript for possible publication. The statistical analysis # refers to different combinations including positive control LPS which stimulates splenocytes, negative control BMS which attenuates stimulation and WA treated groups versus untreated control, we have edited the Figures 2 A and B. We have rectified and substituted the typo errors Con-5 (1 μg/mL) with Con-A (5 μg/mL). We have rectified the splenocyte spelling in Figure 2A
Comment 4: 337-338 our results revealed that WA inhibits the secretion of TNF-α (Figure 3A), IL-6 (Figure 3B), IL-1β (Figure 3C), and nitrite concentration (Figure 3D) in a dose-dependent manner.
In order to demonstrate that the inhibition is dose-dependent, the p values of the comparisons between the different concentrations of WA should be included, and different symbols or bars linking the histograms should be shown. Again, it is not clear the meaning of # (comparison versus presence of only LPS?)
Reply to comment 4: As suggested by the reviewer, we have included the proper symbol which reflects the statistical significance. As suggested by the reviewer, we have indicated # which reflects LPS comparison vs all treatment groups of WA and positive control BMS, and * which reflects untreated control vs LPS.
Comment 5: 341 Additionally, the positive control BMS also showed a significant (# ≤ 0.05) inhibitory effect on cytokine secretion from macrophages compared to the LPS-stimulated and untreated control groups.
I suggest to use two different symbols, for example # to indicate the comparison vs + LPS and * to indicate the comparison versus control (- LPS)
Reply to comment 5: As suggested by the reviewer, we have indicated # which reflects comparison vs +LPS, and * which reflects comparison vs –LPS wherever is required.
Comment 6: Figure 4A, B:
Again, I think that the results should be clearer using different symbols to indicate the statistical significance versus control (* for example) and # versus LPS. Again, if the Authors want to demonstrate that there is a dose-dependent effect, another symbol (for example,§) should be used in the figures to indicate the difference between the higher and the lower WA concentrations, or the information regarding the p values should be added in the text.
Reply to comment 6: As suggested by the reviewer, we have indicated # which reflects comparison vs +LPS, and * which reflects comparison vs –LPS wherever is required.
Comment 7:
Figure 5A, B; Figure 6 B
The Authors should make clear which are the comparisons made, indicate whether # means versus +LPS or versus –LPS, and use two different symbols to indicate the statistical significance versus +LPS and versus –LPS.
Reply to comment 7: As suggested by the reviewer, we have indicated # which reflects comparison vs +LPS and * which reflects comparison vs –LPS wherever is required.
Comment 8: 430-431 Intriguingly, we also observe that WA inhibits Cox-2 and iNOS expression in LPS-stimulated macrophages in a dose-dependent manner (Figure 6 A, B).
It is not clear whether there is a significant dose-dependence in the inhibitory effect of WA on Cox-2 and iNOS expression. To demonstrate that, p values of the comparison between the different concentrations of WA should be included, at least in the text.
Reply to comment 8: As suggested by the reviewer, we have included the p-values in the result section of the manuscript in the above-mentioned lines in red color text.
Comment 9: 505-506 Here, we first demonstrated that WA exhibits dose-dependent LPS-stimulated anti-inflammatory and immunomodulation potential on the peritoneal macrophages and splenocytes derived from BALB/c mice.
There are no statistical significance analyses demonstrating that the effect is dose-dependent, that is, that the effect of the lower dose is different from the effect of the higher dose. The trend is clear from the histograms, but the information on the statistics is missing.
Reply to comment 9: As suggested by the reviewer, we have included statistical significance analysis in figures as well as in the manuscript.
Comment 10: 509 proinflammatory cytokines like NO, TNF-α, IL-1β, and IL-6 both at transcriptional and translational levels.
NO is not a cytokine, although it is a mediator of inflammation.
Reply to comment 10: Thanks for the critical reading of our manuscript. We have removed the word cytokine and replaced it with a mediator of inflammation wherever is required.
Comment 11: 526: However, we observed more than 50 % cell killing when splenocytes were exposed to 2.0 μM of WA for 72 h. See comment above. Killing should be avoided.
Reply to comment 11: As suggested by the reviewer, we have avoided word killing wherever is required in the manuscript.
Comment 12: 546 Upregulation of TNF-α is a major hallmark in the pathophysiology of chronic inflammatory diseases is anti-TNF-α therapy [44]. This sentence is unclear
Reply to comment 12: We have edited the statement and highlighted it in the red color text in the manuscript.
Comment 13: 567-569: we found WA promotes attenuation of iNOS mRNA expression as well as protein expression of iNOS in a dose-dependent manner after LPS-stimulation of macrophages isolated from the peritoneal cavity of BALB/c mice.
See above: to state this it is necessary to add the statistical analyses of the different effects between at least two doses of WA
Reply to comment 13: As suggested by the reviewer, we have incorporated the statistical analysis of above-mentioned results at respective places in the manuscript.
Comment 14: 598-599 Consistent with previous reports, we found a dose-dependent downregulation of p65 mRNA as well as phosphorylation of p-p65 (Ser276, Ser536) protein expression with WA treatment in LPS-stimulated macrophages.
633-634 These results suggest that WA immunosuppresses the spleen-derived LPS-stimulated T and B-cell population in a dose-dependent manner and therefore enables its immunomodulatory effect.
Although I agree that different doses of WA downregulate mRNA, the Authors have not demonstrated a dose-dependent downregulation (see above). It would be necessary to add the statistical analysis showing that there is a difference when comparing at least two doses of WA.
Reply to comment 14: As suggested by the reviewer, we have added statistical analysis of above-mentioned results at respective places in the manuscript.
Minot points:
Comment 1: Some sentences are not clear and need to be rewritten:
23 Inflammation is one of the primary responses and plays a key role in the pathophysiology of various diseases
The sentence seems incomplete. Primary response to what?
Reply to comment 1: As suggested by the reviewer, we have edited the above-mentioned sentences in the manuscript. The edited text is highlighted in red color text.
Comment 2: 33 However, the quantification of B and T cells was performed and flow cytometry.
111 By specifically inhibiting proteasomal chymotrypsin-like activity and protein kinase C [26].
38 NO is not a cytokine
Reply to comment 2: As suggested by the reviewer, we have edited the above-mentioned sentences in the manuscript. The edited text is highlighted in red color text. We have avoided the word cytokine with respect to NO in the manuscript.
Comment 3: Some typos should be corrected:
44 by suppresses
98 immunosuppressant’s [18].
187 with 20 L of MTT The volume unit is wrong. I guess that 20 microL have been added.
188 Centrifuge the plate (1400 x g) for 5min Please rewrite. The plate was centrifuged or after centrifugation…
385 histograms represents
Reply to comment 3: Thanks for the critical reading of our manuscript. We have corrected typo errors at the above-mentioned places as well as other places in the manuscript.

Reviewer 2 Report
You suggest in the introduction that "owing to various side effects... the identification and evaluation of more specific and efficacious anti-inflammatory drugs is seemingly important" which presumably is why you are examining WA, but then later in the introduction you state that "WA interacts with a pletora of molecular targets to exhibit its pharmacological effect" - seemingly contradictory. I think you should remove the statement about specificity and instead state that less toxic and efficacious drugs are needed.
NO is not a cytokine... you mention it in the list of major proinflammatory cytokines; if over produced it is proinflammatory, but it's not a cytokine
In the introduction you mention that "WA restores IkappaB interaction with its binding partner NF-kappaB and promotes inflammation associated signaling pathways"; that's the opposite of what you show here.
line 216 "As mentioned in section 2.4, the collection of peritoneal macrophages..." should be section 2.3
Figure 2 description seemingly interchanges Con-A and Con-5; additionally the text suggests that Con-A was used at 5 µg/ml but the figure legend says 1
Author Response
Reviewer 2:
Comment 1: You suggest in the introduction that "owing to various side effects... the identification and evaluation of more specific and efficacious anti-inflammatory drugs is seemingly important" which presumably is why you are examining WA, but then later in the introduction you state that "WA interacts with a pletora of molecular targets to exhibit its pharmacological effect" - seemingly contradictory. I think you should remove the statement about specificity and instead state that less toxic and efficacious drugs are needed.
Reply to comment 1: As suggested by the reviewer, we have edited the statement in the introduction section of the manuscript.
Comment 2: NO is not a cytokine... you mention it in the list of major proinflammatory cytokines; if over produced it is proinflammatory, but it's not a cytokine
Reply to comment 2: Yes, NO is not a cytokine rather it is a modulator of inflammation. We have edited the sentences in the manuscript.
Comment 3: In the introduction you mention that "WA restores IkappaB interaction with its binding partner NF-kappaB and promotes inflammation associated signaling pathways"; that's the opposite of what you show here.
Reply to comment 3: Thanks for the critical reading of our manuscript. There was a typo error we have substituted prevents word instead of promotes in the above-mentioned sentence in the introduction section of the manuscript.
Comment 4: line 216 "As mentioned in section 2.4, the collection of peritoneal macrophages..." should be section 2.3
Reply to comment 4: Thanks once again for the critical reading of our manuscript. Yes, it is a typo error, we have added 2.3 instead of 2.4 in the 2.4 sub-section of the materials and methods section of the manuscript.
Comment 5: Figure 2 description seemingly interchanges Con-A and Con-5; additionally the text suggests that Con-A was used at 5 µg/ml but the figure legend says 1
Reply to comment 5: We have edited the figure legends Con-A (5 µg/ml).

Reviewer 3 Report
Summary:
Alnuqaydan et al. demonstrate through a series of well-designed experiments that Withaferin A (WA), a plant-derived, steroidal lactone exhibits anti-inflammatory and immunomodulatory effects in LPS-stimulated splenocytes and peritoneal macrophages from BALB/c mice. They found that these effects span the transcriptional and translational points of expression in key pro-inflammatory cytokines as well as upstream and downstream signalling mediators and immunophenotyping.
In my opinion, the study was rigorous, including significant elucidation of the mechanism of action of WA. Conclusions reflect the data presented by the authors.
The manuscript would however benefit from further editing for typographical and grammatical errors in the text, some of which have been highlighted in the comments below – but there are others. Generally, the studies, especially materials and methods, should be reported in the past tense but in many cases in this manuscript, current tenses have been used e.g. lines 181-193.
Can you please describe more clearly how the effect of WA was evaluated with regards to the mice? That is, in vitro or in vivo? Lines 164-172 talks about ‘dosing’ and monitoring the animals but in lines 183-194, it seems splenocytes were isolated the treated externally with WA along with LPS and Con-A. While it is clearer in the later sections of the manuscript what was done, the above referenced sections appear unclear.
See additional specific comments below for details of requested clarifications and corrections.
Comments:
Line 26: Please, change ‘offers’ to ‘offering’
Line 34: change ‘and’ to ‘by’
Line 35: change ‘immunomodulation’ to ‘immunomodulatory’
Line 43: change ‘observe’ to ‘observed’
Line 44: change ‘suppresses’ to ‘suppressor’
Line 50: change ‘supporting’ to ‘support’
Line 62: Please rephrase the following for clarity: ‘is consistent to chemicals…’
Line: 66: Delete comma (,) from the word ‘Besides’
Lines 145-151: Dilution factor for all primary antibodies is the same. Can this be stated in a single sentence instead of separately after each antibody? Similar thing can be done for the secondary antibodies.
Line 165: change ‘fastened’ to ‘fasted’
Line 187: ’20 L’ seems a lot for a 96-well plate! Please, ensure all units are written properly.
Line 195: Were any controls (negative and positive) used in this protocol? Please, include these in the methods. Also, I am curious to know: how many macrophages are generally recovered from this method from a mouse? Many experiments with varying treatments were reported as using these macrophages, with at least three independent experiments. However, the methodology mentions only five mice (line 165).
Line 197: What kind or size of syringe/needle was used?
Line 214: There are inconsistencies in this section with respect to assay plate format and incubation times. Please, reconcile these?
Line 216: Apparently referencing section 2.3 above. Please, verify and correct.
Line 223: ‘As stated in section 2.3’: Please, rephrase or delete this.
Line 243: Provide the centrifugal force used in x g instead of rpm
Lines 292-294: Replace ’48 h compared with 72 h’ with ’48 h and 72 h’. The former sounds as those different trends were obtained for 48 h vs 72 h, but in fact the trends are similar.
Line 297: Figure 1: Describe what was used as control for the assay.
Line 307: BMS control: For information flow, it would help to include the controls, at least initially, where the methods are first describe in the earlier sections.
Line 317: Figure 2: Please, make the scale of the axis the same for panels A and B for consistent representation.
Line 336: Changed ‘conditional media’ to ‘conditioned media’ here and elsewhere
Line 344: Figure 3: Is there a specific reason why y-axis in panel D cannot be given in pg/mL like the rest? If possible, make this consistent for ease of comparison.
Line 345: Consider making Figure 3 legend more concise. Lots of information describing panels A-C are shared information.
Lines 427-428: This sentence appears incomplete; please check and update.
Lines 546-547: Something seems to be missing in this sentence. Please, check and correct.
Author Response
Reviewer 3:
Alnuqaydan et al. demonstrate through a series of well-designed experiments that Withaferin A (WA), a plant-derived, steroidal lactone exhibits anti-inflammatory and immunomodulatory effects in LPS-stimulated splenocytes and peritoneal macrophages from BALB/c mice. They found that these effects span the transcriptional and translational points of expression in key pro-inflammatory cytokines as well as upstream and downstream signalling mediators and immunophenotyping.
Comment 1: In my opinion, the study was rigorous, including significant elucidation of the mechanism of action of WA. Conclusions reflect the data presented by the authors.
Reply to comment 1: Thanks for appreciating our manuscript. We hope our manuscript will be accepted for publication.
Comment 2: The manuscript would however benefit from further editing for typographical and grammatical errors in the text, some of which have been highlighted in the comments below – but there are others. Generally, the studies, especially materials and methods, should be reported in the past tense but in many cases in this manuscript, current tenses have been used e.g. lines 181-193.
Reply to comment 2: As suggested by the reviewer, we have edited the manuscript to rectify typographical and grammatical errors in the entire manuscript. In the materials and methods section as suggested by the reviewer, we have rewritten the text sentences in the past tense.
Comment 3: Can you please describe more clearly how the effect of WA was evaluated with regards to the mice? That is, in vitro or in vivo? Lines 164-172 talks about ‘dosing’ and monitoring the animals but in lines 183-194, it seems splenocytes were isolated the treated externally with WA along with LPS and Con-A. While it is clearer in the later sections of the manuscript what was done, the above referenced sections appear unclear.
Reply to comment 3: We have clearly mentioned in the materials and methods section under subheading 2.1, para 2 that The current study was conducted on immune cells ( macrophages and splenocytes) isolated from BALB/c (male) mice and then cultured under aseptic conditions to evaluate the anti-inflammatory and immunomodulatory effects of WA. Thanks for the critical reading of our manuscript, we have removed lines 164-172 from the materials and methods section which is not related to the manuscript.
See additional specific comments below for details of requested clarifications and corrections.
Additional Comments:
Comment 1: Line 26: Please, change ‘offers’ to ‘offering’
Line 34: change ‘and’ to ‘by’
Line 35: change ‘immunomodulation’ to ‘immunomodulatory’
Line 43: change ‘observe’ to ‘observed’
Line 44: change ‘suppresses’ to ‘suppressor’
Line 50: change ‘supporting’ to ‘support’
Line 62: Please rephrase the following for clarity: ‘is consistent to chemicals…’
Line: 66: Delete comma (,) from the word ‘Besides’
Reply to comment 1: As suggested by the reviewer, we have edited the manuscript as per the above-mentioned suggestions.
Comment 2: Lines 145-151: Dilution factor for all primary antibodies is the same. Can this be stated in a single sentence instead of separately after each antibody? Similar thing can be done for the secondary antibodies.
Line 165: change ‘fastened’ to ‘fasted’
Line 187: ’20 L’ seems a lot for a 96-well plate! Please, ensure all units are written properly.
Reply to comment 2: As suggested by the reviewer, we have made a single sentence to present the dilution factor of primary antibodies used in immunoblotting. We have substituted fasted instead of fastened and have written units properly.
Comment 3: Line 195: Were any controls (negative and positive) used in this protocol? Please, include these in the methods. Also, I am curious to know: how many macrophages are generally recovered from this method from a mouse? Many experiments with varying treatments were reported as using these macrophages, with at least three independent experiments. However, the methodology mentions only five mice (line 165).
Reply to comment 3: As suggested by the reviewer, we have added negative or positive control such as LPS and BMS in the materials and methods section. Usually, we collect 3-4 ×105 cells per mouse. The paragraph which mentions 5 mice was wrongly incorporated here and have removed. At least three independent experiments were conducted in macrophages isolated from the peritoneal cavity of mice.
Comment 4: Line 197: What kind or size of syringe/needle was used?
Reply to Comment 4: A 10 ml syringe with a 20-23 G beveled needle is used to collect peritoneal macrophages.
Comment 5: Line 214: There are inconsistencies in this section with respect to assay plate format and incubation times. Please, reconcile these?
Reply to comment 5: As suggested by the reviewer, we have edited the section and made it more consistent and meaningful.
Comment 6: Line 216: Apparently referencing section 2.3 above. Please, verify and correct.
Line 223: ‘As stated in section 2.3’: Please, rephrase or delete this.
Line 243: Provide the centrifugal force used in x g instead of rpm
Reply to comment 6: We have corrected the referencing section 2.3 and have provided centrifugal force into x g instead of rpm
Comment 7: Lines 292-294: Replace ’48 h compared with 72 h’ with ’48 h and 72 h’. The former sounds as those different trends were obtained for 48 h vs 72 h, but in fact the trends are similar.
Reply to comment 7: As per the reviewer’s suggestions. We have rectified it in 48 h and 72 h.
Comment 8: Line 297: Figure 1: Describe what was used as control for the assay.
Reply to comment 8: In figure 1, the control used is untreated splenocytes.
Comment 9: Line 307: BMS control: For information flow, it would help to include the controls, at least initially, where the methods are first describe in the earlier sections.
Reply to comment 9: As suggested by the reviewer, we have incorporated BMS as a BMS control
Comment 10: Line 317: Figure 2: Please, make the scale of the axis the same for panels A and B for consistent representation.
Line 336: Changed ‘conditional media’ to ‘conditioned media’ here and elsewhere
Reply to comment 10: As suggested by the reviewer, we have made a single scale in both figures 2A and B for consistent representation. We have substituted conditioned media instead of conditional media in line 344.
Comment 11: Line 344: Figure 3: Is there a specific reason why y-axis in panel D cannot be given in pg/mL like the rest? If possible, make this consistent for ease of comparison.
Line 345: Consider making Figure 3 legend more concise. Lots of information describing panels A-C is shared information.
Reply to comment 11: Since the common unit for the measurement of Nitrite concentration is µM instead of pg/mL in other cytokines. That is the reason we put µM units for nitrite measurement. Regarding the legend concise of figure 3, as suggested by the reviewer, we concise the legend of figure 3 to make it more meaningful.
Comment 12: Lines 427-428: This sentence appears incomplete; please check and update.
Lines 546-547: Something seems to be missing in this sentence. Please, check and correct.
Reply to comment 12: As suggested by the reviewer, we have edited the above-mentioned sentences to make them more catchy and meaningful.
